# A Cross-Regulatory Framework for Balancing Compliance Obligations in Decentralised Settings via Semantic Actor, Rights and Obligations Alignment⋆

Gabriela Kurteva[1,*], Harshvardhan J. Pandit[1], Dave Lewis[1] and Edoardo Celeste[2]

[1]*ADAPT Centre, Trinity College Dublin, College Green, Dublin 2, Ireland*
[2]*School of Law and Government, Dublin City University, Dublin 9, Ireland*

## Abstract

Solid brings data sovereignty back to individuals, but with the emergence of AI and EU data regulations, a concern with its technical implementation may arise. In Solid, enforcement of usage control (i.e. what must be satisfied to allow 'read', 'write' operations) is limited and decentralisation challenges identifying who holds a role, what the role is, and the legal accountability one must hold. The lack of case laws helping to translate legal principles to fit decentralised environments exacerbates this legal uncertainty. The mainstream of centralised architectures and the societal lack of experience with decentralised technologies compound to this. Transforming EU governance requirements into a usage control policy with a legal ontology Solid Pods can interpret may address the challenge.

## Keywords

Decentralised Data Sharing, Ontologies, Cross-regulatory Compliance, Solid

## 1. Introduction

The General Data Protection Regulation (GDPR) [1], Data Act [2] and Data Governance Act (DGA) [3] impose obligations founded on data. Different roles, considering who can move, delete or provide access to data, decide the type needed and how to process it, are introduced. In centralisation, a Data Controller controls the application and server, whereas in Solid, a user controls data access, a Pod maintains storage, and an application determines processing purposes. This fragmentation does not necessarily make Solid and the GDPR conflicting as a Data Controller hosting a Solid server might have a Data Processor managing partially its functionalities. The primary alignment of Solid with the GDPR compliance through access controls makes it well set up for authorization processes, but prone to auditing deficits. This may owe to the absent standardised data storage format and a logging layer, recording access events, legal basis, and purpose, also resulting in neglect of DGA's transparency requirements [4]. Further, Solid cannot understand what roles an actor holds [5], and only grants reading, writing and appending permissions. It cannot translate EU laws into interpretable requirements, which may pose misalignment during their enforcement. This technical and operationalization challenge can put accountability, transparency and rights at risk [6]. The lack of semantic documentation for compliance analysis and auditing metadata within data sharing processes also limit protections when access is violating and impose human-led audits. Nonetheless, developments in using legally-aligned policy languages [7] bring decentralised systems closer to legal compliance. For example, work for valid GDPR consent allows Data Controllers to declare their data processing practices, while supporting Data Subjects in expressing data sharing preferences. Yet, vagueness in encoding legal logic in Solid's behaviour and normative friction if a decentralised actor holds various roles with overlapping obligations may persist due to Solid's nature. Resulting uncertainties of which obligation should supersede may further permit data transaction breaches of regulation(s) the system cannot locate. Therefore, the proposal questions:
*"To what extent can semantic representations align concepts for actors, rights and obligations from European*

*The 4th Solid Symposium, April 30–May 01, 2026, London, UK*
⋆Corresponding author.
✉ gabriela.kurteva@adaptcentre.ie (G. Kurteva)
🆔 0000-0009-5357-575 (G. Kurteva)

*regulations and support cross-regulatory compliance and data governance in decentralised systems?".*

**Scope of research and Limitations:**    The base for promoting governance in decentralized environments is defined by **(i) Actor Scope**: individuals, functional, infrastructure actors (e.g., controllers); **(ii) Legal Scope**: cross-jurisdiction compliance across GDPR, DGA, Data Act; **(iii) Accountability:** centering semantic mapping and obligation-based policies, grounded in expert validation, for semantic logic and legal context correspondence. Particular parameters, although not considered, can drive future work: **(i) Behaviour compliance:** semantic framework use to align human factors in compliance decisions with legal metadata and enable interface development, bringing social realities with data sovereignty closer; **(ii) Cross-jurisdictional Scope:** adoption of non-EU regulations may reduce regulatory obsolescence, making Solid compliant with global legal obligations.

## 2. Methodology

To assess if semantic techniques can bridge decentralized data sharing and regulatory demands, **Phase 1** seeks to understand what may make Pod's compliance difficult. Iterative cross-regulatory alignments of actor-rights-obligations in RDF format using vocabularies such as Data Privacy Vocabulary (DPV) [8] is also involved, potentially extending DPV for Legal Role Assertions linked to Solid's technical actors and semantic privacy notices [9]. The phase looks to assess how well Solid understands which legal rules apply by aligning extracted concepts with technical identifiers and roles. Legal expert inclusion (e.g., legal scholars), including in Phase 2, can validate if the regulation-targeted taxonomies reflect the nuanced concept meanings. **Phase 2** seeks to harmonize them into an ontology, determining its feasibility to support their attachment to data resources. Policy languages within the semantic layer, where data requests meet and are verified for compliance, can also embed rules as permission-prohibition-obligation triples. The phase seeks to evaluate if overlapping obligations can also be mapped with a transaction to enable verification against law requirements. Open Digital Rights Language (ODRL) [10] can translate law's conditional behaviour in a structure decentralised systems accept as requirements. **Phase 3** explores SHACL[1] to see if the pre-defined shapes of legal requirements and the semantic representation fit, and confirm metadata availability within incoming requests for legal analysis. SHACL can check if an actor holds a correct 'role' to exercise rights and if a legal term, mapped to a technical event, can clarify how an action is authorised. It considers SPARQL[2] to create receipts of these actions.

## 3. Conclusions

This research proposes a semantic framework for harmonising legal concepts across EU Data regulations towards Solid to bring discussions on how legislative revisions and compliance reproducibility might be approached. For regulators and companies, the research aspires to demonstrate how EU laws, through guideline documentation, and cross-regulation compliance may be addressed. For individuals, it searches how formalising responsibilities can reduce legal uncertainty and improve data interoperability for parties concerned. In the future, this work may facilitate better enforcement of legal accountability and product history, when data travels between entities in circular economy (e.g., Digital Product Passport).

## Acknowledgments

This proposal is funded by the European Union's Horizon Europe programme under the Marie Skłodowska-Curie Actions project HARNESS (grant agreement No.101169409). The ADAPT Centre for Digital Media Technology is funded by Research Ireland through the Research Centres Programme and is co-funded under the European Regional Development Fund through Grant 13/RC/2106_P2.

---

[1]https://www.w3.org/TR/shacl/
[2]https://www.w3.org/TR/sparql11-query/

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
