# OpenReview forum: "A Cross-Regulatory Framework for Balancing Compliance Obligations in Decentralised Settings via Semantic Actor, Rights and Obligations Alignment"
_SolidProject.org/SoSy/2026/Privacy_Session — SoSy2026-Privacy Abstract_

### Official Review · ~Cesar_Augusto_Fontanillo_López1 · 2026-02-27
**The submission remains underdeveloped**

**Rating:** 4
**Confidence:** 3

**Review:**

The submission attempts to investigate how to align Solid with legal data protection requirements by translating legal roles, rights, and obligations into machine-interpretable semantic representations that can be enforced and audited. However, it seems that the legal problems that the submission raises are not well explained and, from time to time, seem to be a bit assumptive. Formulations about legal problems remain rather abstract and are not, in all cases, grounded in literature and illustrative examples. It would be advisable if the submission highlighted a small set of concrete data protection challenges and were more specific about how the research would address them.

The section on the scope of research and limitations seems to be underdeveloped. It does not come across clearly what the limitations are in relation to the factors that are enumerated, nor what the scope of the research would be. There seems to be a framing problem, which hinders the possibility of fully understanding what the authors aim to actually do in relation to the research scope and what limitations it will have. Similarly, the methodology section does not allow one to fully understand how the three phases are supposed to jointly answer the main research question and how concretely they will achieve this objective. It does not talk either about how the solution that is sought is to be evaluated or validated in relation to legal requirements. Lastly, the submission jumps directly to the conclusions, without further ado.

It seems that the submission is underdeveloped. The motivation behind the research question remains insufficient and not well-explained or grounded. A literature review (even a short one) of previous and related work in the area is lacking to justify the research gap and relevance of the proposed research. The section on the scope and limitations of the research remains equally underdeveloped, and the methodology section could be improved. The work may potentially have research value, but as it stands, it requires further explanation and refinement, in particular to satisfy the criteria of quality, clarity, originality, and significance. I am not in a position to evaluate how well they are satisfied with the available information.

---

### Official Review · ~Arianna_Rossi1 · 2026-03-02
**Well-articulated research proposal**

**Rating:** 9
**Confidence:** 4

**Review:**

Drawing from a clear list of regulatory-related limitations of Solid, this paper proposes a well-articulated research plan for addressing them. The analysis of existing issues concerning compliance with EU data regulatory frameworks is solid and concise. Expanding beyond the compliance with the GDPR to include compliance with the Data Act and the DGA is a recognized need with tangible scientific and societal implications. The methodology is sketched out at a high level but still manages to explain the various research phases and the methods that will be employed.

The proposal would benefit from a clearer articulation of the expected results and the concrete outputs of the research. The link between the research plan and the expected usefulness for companies, regulators and individuals needs to be more explicitly established.

Moreover, given the lack of case law identified by the authors and the relativity which hampers legal certainty that existing case law sometimes can introduce (e.g., the redefinition of personal data in Case C-413/23 P. that has been now introduced in the Digital Omnibus proposal), how exactly legal experts will participate in this research and how reliable knowledge (“nuanced concept meanings”) will be extracted from their expertise needs to be carefully delineated, from identifying the appropriate kind of expertise to be included to designing the actual study. The mapping and interrelation of EU data regulations are very complex and open-ended, thus formalizing them may be challenging because of this lack of established knowledge.

In sum, this proposal is well-formulated and justified by existing gaps, thus it carries high potential in terms of relevance. However, feasibility may be hampered by ever-evolving legal knowledge (but lessons may be drawn from other successful projects), which needs to be carefully addressed.

---

### Decision · Program_Chairs · 2026-03-09

Accept (Abstract)